# Time-Dependent Molecular Motifs of Pulmonary Fibrogenesis in COVID-19

**DOI:** 10.3390/ijms23031583

**Published:** 2022-01-29

**Authors:** Jan C. Kamp, Lavinia Neubert, Maximilian Ackermann, Helge Stark, Christopher Werlein, Jan Fuge, Axel Haverich, Alexandar Tzankov, Konrad Steinestel, Johannes Friemann, Peter Boor, Klaus Junker, Marius M. Hoeper, Tobias Welte, Florian Laenger, Mark P. Kuehnel, Danny D. Jonigk

**Affiliations:** 1Department of Respiratory Medicine, Hannover Medical School, 30625 Hannover, Germany; fuge.jan@mh-hannover.de (J.F.); Hoeper.Marius@mh-hannover.de (M.M.H.); Welte.Tobias@mh-hannover.de (T.W.); 2Biomedical Research in Endstage and Obstructive Lung Disease Hannover (BREATH), German Center for Lung Research (DZL), 30625 Hannover, Germany; Neubert.Lavinia@mh-hannover.de (L.N.); Stark.Helge@mh-hannover.de (H.S.); Werlein.Christopher@mh-hannover.de (C.W.); Haverich.Axel@mh-hannover.de (A.H.); Laenger.Florian@mh-hannover.de (F.L.); Kuehnel.Mark@mh-hannover.de (M.P.K.); Jonigk.Danny@mh-hannover.de (D.D.J.); 3Institute of Pathology, Hannover Medical School, 30625 Hannover, Germany; 4Institute of Pathology and Department of Molecular Pathology, Helios University Clinic Wuppertal, University of Witten-Herdecke, 42283 Wuppertal, Germany; maximilian.ackermann@uni-mainz.de; 5Institute of Functional and Clinical Anatomy, University Medical Center of the Johannes Gutenberg-University Mainz, 55122 Mainz, Germany; 6Department of Cardiothoracic, Transplant and Vascular Surgery, Hannover Medical School, 30625 Hannover, Germany; 7Institute of Medical Genetics and Pathology, University Hospital Basel, 4031 Basel, Switzerland; Alexandar.Tzankov@usb.ch; 8Institute of Pathology and Molecular Pathology, Bundeswehrkrankenhaus Ulm, 89081 Ulm, Germany; konradsteinestel@bundeswehr.org; 9Institute of Pathology, Märkische Kliniken GmbH, Klinikum Lüdenscheid, 58515 Lüdenscheid, Germany; johannes.friemann1@uk-koeln.de; 10Institute of Pathology and Department of Nephrology, RWTH University of Aachen, 52062 Aachen, Germany; pboor@ukaachen.de; 11Institute of Pathology, Bremen Central Hospital, 28177 Bremen, Germany; klaus.junker@klinikum-bremen-mitte.de

**Keywords:** COVID-19, SARS-CoV-2, pulmonary fibrosis, inflammation

## Abstract

(1) Background: In COVID-19 survivors there is an increased prevalence of pulmonary fibrosis of which the underlying molecular mechanisms are poorly understood; (2) Methods: In this multicentric study, n = 12 patients who succumbed to COVID-19 due to progressive respiratory failure were assigned to an early and late group (death within ≤7 and >7 days of hospitalization, respectively) and compared to n = 11 healthy controls; mRNA and protein expression as well as biological pathway analysis were performed to gain insights into the evolution of pulmonary fibrogenesis in COVID-19; (3) Results: Median duration of hospitalization until death was 3 (IQR25-75, 3–3.75) and 14 (12.5–14) days in the early and late group, respectively. Fifty-eight out of 770 analyzed genes showed a significantly altered expression signature in COVID-19 compared to controls in a time-dependent manner. The entire study group showed an increased expression of *BST2* and *IL1R1*, independent of hospitalization time. In the early group there was increased activity of inflammation-related genes and pathways, while fibrosis-related genes (particularly *PDGFRB*) and pathways dominated in the late group; (4) Conclusions: After the first week of hospitalization, there is a shift from pro-inflammatory to fibrogenic activity in severe COVID-19. *IL1R1* and *PDGFRB* may serve as potential therapeutic targets in future studies.

## 1. Introduction

Since the first documented infection of severe acute respiratory syndrome coronavirus 2 (SARS-CoV-2) in Wuhan, China in December 2019, more than 279 million confirmed cases have been documented worldwide resulting in more than 5 million cumulative deaths [WHO Coronavirus (COVID-19) Dashboard, last accessed on 28 December 2021; Internet: https://covid19.who.int/]. Over the last two years, a vast number of studies have been performed to identify associated clinical and morphological features of coronavirus disease 2019 (COVID-19) as well as the underlying molecular mechanisms.

Most individuals infected by SARS-CoV-2 present with mild to moderate symptoms such as a dry cough and fever and recover within two weeks. However, approximately 20% of patients develop a more severe inflammatory response after the initial disease phase, ultimately 5–7% requiring intensive care treatment [1]. The overall mortality rate of COVID-19 is reported to be about 2%, while significantly higher mortality has been reported depending on age, comorbidities, and vaccination status [2,3] as well as in distinct subgroups such as solid organ transplant recipients [4,5].

A considerable number of COVID-19 survivors continue to suffer from persistent symptoms such as fatigue, dyspnea/abnormal breathing or chest pain, summed up under the umbrella term “long-COVID syndrome”. Recently, a large retrospective cohort study revealed a presence of at least one long-COVID symptom in more than one-third of 273.618 patients between three to six months after surviving a SARS-CoV-2 infection [6]. Moreover, persistent radiological alterations were found in a large number of COVID-19 survivors. In patients with ongoing respiratory symptoms three months after infection, the presence of ground-glass opacities and bands was reported in 48% and 37%, respectively, as well as signs indicative of established fibrosis, such as volume loss and traction bronchiectasis in 12% [7]. In addition, pulmonary alterations that can be found on a histological level in patients who succumbed to COVID-19, include diffuse alveolar damage and acute fibrinous and organizing pneumonia [8,9]; two alterations that are known to potentially induce pulmonary parenchymal fibrosis.

Progressive pulmonary fibrosis is a well-known and largely described sequela of acute respiratory distress syndrome (ARDS) [10]. Concerning ARDS in the context of respiratory infections, pulmonary fibrosis is much more frequently resulting from bacterial, than from viral infections of the lower respiratory tract. However, bacterial superinfections can be the result of initial viral infections, e.g., in the context of influenza [11]. Typical, but unspecific, histological hallmarks of ARDS, termed diffuse alveolar damage, are hyaline membranes, fibrinous exudates and intra-alveolar edema. However, COVID-19-related ARDS is characterized by further disease-specific histological hallmarks: (i) a predominantly angiocentric inflammation with microangiopathy and increasing prevalence of intussusceptive pillar formations over time (>7–10 days) indicating the initiation of an aberrant “intussusceptive” neo-angiogenesis [12]; (ii) a high prevalence of thrombi in the small arterioles and capillaries; (iii) a tight interaction between thrombi and adjacent myofibroblasts. These characteristics were also indicated at the gene expression level in COVID-19 autopsy samples showing a distinct and specific pattern of angiogenesis-related genes different from the pattern seen in influenza autopsy samples [12]. The evolution of these alterations is based on a complex interplay between endothelial cells and myofibroblasts termed as the “fibrovascular interface” [13], representing a functional interconnection between the mechanisms of angiogenesis and fibrogenesis. Gene expression analysis has proven to be a useful way of characterizing the complexity of the underlying processes by comparison of similar, yet different entities. In COVID-19, much insight into the pathomechanisms of the disease has been achieved by systematic analysis of gene expression profiles in all kinds of available biomaterials, such as homogenization of whole lung lobes, nasopharyngeal swabs, peripheral blood mononuclear cells, and broncho-alveolar lavage fluids using RNAseq and Nanostring analysis [14,15,16,17]. Here, samples from healthy vs. mildly ill vs. severely ill patients were used to describe the evolution of the disease from mild to severe, using a correlative analysis approach. In previous studies, we worked out the central significance of angiogenesis and fibrosing pulmonary injury patterns in the context of COVID-19. The functional relevance of the fibrovascular interface for fibrogenesis is still unknown, but in the light of recent advances in COVID-19 might be underestimated [18]. Various fibrosing injury patterns have been found in autopsy lung samples from patients who succumbed to COVID-19 [19,20]. Recently, similarities were found between macrophage populations in COVID-19 and idiopathic pulmonary fibrosis. Moreover, exposure of human monocytes to SARS-CoV-2 was shown to induce a profibrotic phenotype in vitro suggesting a role of pro-fibrotic macrophages in the development of fibroproliferative acute respiratory distress syndrome in COVID-19 [21].

Taking into account these findings, it appears logical to elucidate the molecular processes that contribute to the early stages of pulmonary fibrogenesis in patients with COVID-19, to identify potential therapeutic targets for the prevention, or impeding of pulmonary fibrosis and its concomitant consequences. As intussusceptive angiogenesis manifests very early in COVID-19 lung injury, we performed comprehensive fibrosis-specific mRNA and protein expression analyses in the lungs of patients who succumbed during early (<7 days of hospitalization) or late (>7 days of hospitalization) COVID-19 and compared these to healthy controls.

## 2. Results

### 2.1. Clinical Information

The study group comprised n = 12 patients from six hospital sites from Germany and Switzerland who were hospitalized due to COVID-19 and died of progressive respiratory failure. N = 6 patients each were assigned to the “early group” and the “late group”, respectively, with a median duration of hospitalization time until death of 3 (IQR25-75, 3–3.75) and 14 (12.5–14) days, respectively. Mechanical ventilation was applied in n = 4 and n = 5 patients of the early and late groups, respectively. Only two patients, both in the late cohort, had documented comorbidities. One patient suffered from moderate obesity and one patient had a documented abstinent nicotine dependence. No patients suffered from pulmonary comorbidities, particularly no patient had a history of fibrosing lung diseases. More details are depicted in the Appendix A.

### 2.2. mRNA Expression

As depicted in Figure 1, we found 58 genes of a total of 770 genes analyzed differentially expressed in COVID-19 compared to the control group. Of these, nine showed differential expression only in the “early group”, 40 only in the “late group” and nine in both groups. Box plots of all genes with a significantly increased expression on the mRNA level are depicted in Appendix A. In all patients who succumbed to COVID-19, we found increased expression of bone marrow stromal cell antigen 2 (*BST2*) and interleukin 1 receptor type 1 (*IL1R1*) independent of hospitalization time. In patients who died during the first week of hospitalization, increased expression was found for ADAM metallopeptidase domain 17 (*ADAM17*), complement component 1 subcomponent S (*C1S*), DNA damage-inducible transcript 3 (*DDIT3*), and serpin family G member 1 (*SERPING1*). In patients who died later than day seven of hospitalization, increased mRNA expression was found for ADAM metallopeptidase domain 9 (*ADAM9*), BCL2 associated X apoptosis regulator (*BAX*), caspase 4 (*CASP4*), cytoskeleton-associated protein 4 (*CKAP4*), several collagen subtypes (i.e., *1A1*, *1A2*, *3A1*, *4A1*, *5A1*, and *6A3*), casein kinase 1 epsilon (*CSNK1E*), hypoxia-inducible factor 1 alpha (*HIF1A*), major histocompatibility complex class IA and IB (*HLA-A/B*), insulin-like growth factor 1 (IGF1), integrin subunit beta 1 (*ITGB1*), matrix metalloproteinase 1 and 14 (*MMP1/14*), platelet-derived growth factor receptor beta (*PDGFRB*), phosphoglycerate kinase 1 (*PGK1*), serpin family F member 1 (*SERPINF1*, also known as pigment epithelium-derived factor, *PEDF*), serpin family H member 1 (*SERPINH1*), and thrombospondin 2 (*THBS2*). The mRNA expression raw data and further information of gene functions are provided in the Appendix A.

### 2.3. Functional Analysis

Analysis of biological pathways based on the gene expression profiles was performed using gene-pathway associations supplied by Nanostring as well as the GeneOntology database. This analysis revealed specific functional differences in early versus late COVID-19 patients. In general, COVID-19 samples were characterized by increased activity of several inflammatory and fibrosis-related pathways in a specific time-dependent manner as depicted in Figure 2 and Appendix A. In patients who succumbed to COVID-19 during the first week of hospitalization, we predominantly found an up-regulation of inflammatory pathways such as interferon signaling pathways, pathways for antigen presentation and T-cell differentiation as well as cytokine, NF-kappa B and NOD-like receptor signaling pathways, while in patients who died later, fibrosis-related pathways such as metabolism and organization of collagens, as well as remodeling by matrix metalloproteinases and epithelial-to-mesenchymal transition pathways were markedly pronounced with concomitant down-regulation of inflammatory pathways. More details on the shift from inflammation- to fibrosis-related pathways are shown in Appendix A.

### 2.4. Protein Expression

Based on the detected mRNA expression profiles, the presence of ITGB1, MMP14, THBS2, COL1, and PDGFRB were compartment-specifically assessed on the protein level via immunohistochemistry. As depicted in Figure 3, Figure 4 and Figure 5 as well as in the Appendix A, all five of these proteins showed an increased expression in the late compared to the early group. Expression of MMP14 was found on type I and II pneumocytes, macrophages and lymphocytes in the early group as well as in fibroblast-like cells in the late group. ITGB1 was found on endothelial cells (especially capillaries) and smooth muscle cells (particularly within the bronchial wall) in the early group as well as on macrophages, type I and II pneumocytes, and fibroblast-like cells in the late group. COL1 was found within the perivascular soft tissue (and a subset of macrophages) in the early group and pronounced but in the same localization in the late group. THBS2 was found on smooth muscle cells and macrophages in both groups, mildly pronounced in the late group. PDGFRB was found in macrophages, type I and II pneumocytes, and in smooth muscle cells, mainly in the late group. Staining of healthy control samples using these five antibodies was performed but is not depicted within this manuscript. The localization of proteins in control samples was the same as in COVID-19 lungs but, in part, staining intensity differed what might reflect the difference in tissue integrity between partially autolyzed autopsy samples and immediately worked-up control tissue.

Further immunostaining was performed using antibodies for all collagen subtypes that showed increased activity on the mRNA level only in the late group, with the aim to discriminate their amount and distribution patterns. As collagens are generally present in lung tissue, immunostaining was also performed using healthy samples from lungs that were donated but not used for transplantation as controls, as presented in Appendix A. COL1 showed a specific enhancement within the perivascular stromal tissue with a distinct increase in the late COVID-19 group. COL3 and COL4 were predominantly enriched within the internal and external elastic laminas of the pulmonary artery branches and the alveolar basement membranes while COL5 and COL6 showed a specific enhancement within the media and adventitial layer of the pulmonary artery branches, the surrounding perivascular stromal tissue, and the septal connective tissue. All collagen subtypes showed a markedly increased expression in the late COVID-19 group.

## 3. Discussion

Post COVID-19 respiratory impairment—as part of the “long COVID” spectrum—has been linked to aberrant fibrotic remodeling. To address this, we analyzed the time-dependent mRNA and protein expression profiles of the lungs of 12 patients who succumbed to COVID-19. We assessed the gene expression profiles relative to the hospitalization time, as well as to age and sex-matched healthy controls. One major finding was the predominance of inflammatory pathways within the first week of hospitalization and the subsequent shift to predominantly fibrosis-related pathways in week two defined by the significant expression profiles of 58 genes accompanied by increased protein expression of several collagen subtypes with increasing hospitalization time. 

Interestingly, upon the 58 differentially expressed genes, only two genes that showed increased activity were characteristic for all COVID-19 lungs independent of hospitalization time: *IL1R1* and *BST2*. IL1R1 is one of the most important mediators involved in a wide range of immune and inflammatory responses induced by cytokines and associated with several pro-inflammatory signaling pathways, e.g., nuclear factor κB (NF-κB) and mitogen-activated protein kinase (MAPK) signaling [22,23]. In a former study by Calverley et al., patients with COPD were treated with a human immunoglobulin G2 monoclonal antibody targeting IL1R1 called MEDI8968 (ClinicalTrials.gov identifier: NCT01448850) to prevent acute exacerbations [24]. The overall safety profile of MEDI8968 was rated as acceptable; however, there were no statistically significant improvements in acute exacerbation rate, lung function or quality of life. As IL1R1 appears to play a role in COVID-19, MEDI8968 might be a future therapeutic option in patients with severe COVID-19 to prevent overwhelming inflammatory host responses. In addition, the increased expression of IL1R1 supports the use of the interleukin 1 receptor antagonist anakinra. Several studies addressed the use of anakinra in the context of severe COVID-19 and a reduction of the need for invasive mechanical ventilation and mortality without severe side-effects could be shown [25,26,27]. However, randomized controlled studies are needed to confirm these results. BST2 is an interferon-induced type-II membrane protein and a major host restriction factor that plays an important role in the innate immune response to viral infections [28,29]. BST2 inhibits the release of several enveloped viruses including coronaviridae by tethering the virions to the membranes of infected cells [30].

The gene expression and functional analysis of the early vs. late COVID-19 cases revealed significant differences with increased levels of *ADAM17*, *C1S*, *SERPING1*, and *DDIT3* in the early group. *ADAM17* encodes a member of a large family of disintegrin and metalloprotease domains involved in several biological processes such as cell-cell and cell-matrix interactions as well as processing variant substrates, e.g., cytokines and growth factor receptors [31]. In addition, ADAM17 is involved in the Notch signaling pathway, which plays an important role in the differentiation and activation of innate and adaptive immune cells and is therefore crucial for the cytokine storm formation in patients with severe COVID-19 [32,33]. Recent works also suggested a role of ADAM17 in the SARS-CoV-2 infection via its capability of cleavage of the angiotensin-converting enzyme type 2 (ACE2) receptor [34]. The serine protease C1s is a constituent of the human complement factor C1 and contributes to the activation of the so-called classical pathway [35] while SERPING1 serves as its inhibitor and therefore downregulates complement activation. *DDIT3* encodes a multifunctional transcription factor involved in variant cell stress response signaling pathways and in the induction of caspase-4/11, the latter indirectly inducing inflammatory responses via activation of pro-interleukin 1B (pro-IL1B) to mature IL1B [36].

We could identify two major findings discriminating the early from the late COVID-19 cases with regard to potential therapies.

First, the increased expression of *PDGFRB* both on the mRNA and on the protein level in those patients who succumbed in the second week of hospital stay or later. PDGFRB is a well-described tyrosine kinase that plays a major role in the development of pulmonary fibrosis [37]. Its ligand, platelet-derived growth factor subunit B (PDGFB), acts as a strong chemoattractant for fibrocytes. PDGFRB is targeted by several tyrosine kinase inhibitors such as nintedanib, imatinib, and sunitinib. Of these, nintedanib represents one of only two currently available antifibrotic agents [38,39,40]. The increased expression of *PDGFRB* suggests that patients with severe COVID-19 might benefit from an early nintedanib therapy to prevent pulmonary fibrosis following SARS-CoV-2 infection. Nintedanib could significantly reduce vascular proliferation and normalized the distorted microvascular architecture in a lung fibrosis model [41]. In a recent study with 30 critically ill COVID-19 patients by Umemura et al., nintedanib showed a potential to reduce the time of mechanical ventilation while mortality remained unchanged [42]. To further elucidate the therapeutic potential of nintedanib in this context, there is an ongoing clinical trial by the manufacturer (ClinicalTrials.gov identifier: NCT04541680, accessed on 27 January 2022) with an estimated enrollment of 250 patients. Therein, nintedanib therapy is initiated in the early disease phase and administered twice daily over one year with an aim to inhibit the activation of mesenchymal cells and the progression of virus-induced pulmonary fibrosis. Of course, the late group of this study still represents a subacute patient population, and predictions on long-term nintedanib treatment are not possible based on this data. However, an early nintedanib treatment might prevent the transition from initial inflammation to early stages of fibrogenesis. 

Second, the late group displayed an increased activity of HIF1A. HIF1A encodes the alpha unit of a transcription factor that regulates the cellular and systemic homeostatic response to hypoxia by activating several genes many of which are involved in energy metabolism, angiogenesis, and apoptosis [43]. Thus, HIF1A is crucial for many developmental processes such as embryonic vasculogenesis and neo-angiogenesis as well as in multifarious clinical conditions associated with hypoxia. Interestingly, in a recent work, Serebrovska et al. could show that HIF1A reduces the presence of ACE2 on the cell surface upon SARS-CoV-2 infection via an up-regulation of ADAM17 [44]. As discussed above, ADAM17 promotes cleavage of ACE2 resulting in an aggravated cell entry for SARS-CoV-2 [45,46]. Taken together, these findings suggest a prominent role of HIF1A in COVID-19 not only as a key molecule for the management of tissue hypoxia but also as a coordinator of infection control and controller of SARS-CoV-2 invasiveness. Moreover, HIF1A is suspected to be a driver of a special mechanism of neovascularization termed intussusceptive angiogenesis [47].

Intussusceptive angiogenesis, also known as splitting angiogenesis, is, in contrast to sprouting angiogenesis, characterized by the division of an existing blood vessel into two new lumens, a highly dynamic process that occurs within minutes to hours [48,49]. Recently, we analyzed the molecular mechanisms behind intussusceptive angiogenesis in COVID-19 hearts [50]. Therein, we suggested that the endothelial damage, caused by COVID-19, leads to impaired microcirculation, thrombogenesis and a reactive up-regulation of adhesion-molecules as well as activation of SDF-1/CXCR4 signaling. This altered microenvironment, in turn, attracts CD11+/TIE2+ macrophages, which are activated by increased angiopoietin-1 (ANGPT1) levels and adhere to the endothelial surface, thereby inducing a stretching of these endothelial cells and resulting in intussusceptive pillar formation. Subsequently, blood-borne angiogenetic progenitor cells are incorporated into these pillars resulting in a vessel duplication. Thus, intussusceptive angiogenesis leads to aberrant expansion of the vascular plexus and progressive remodeling of the fibrovascular interface.

In our late patient group, we found up-regulation of the serine protease inhibitor-coding gene *SERPINF1* and *THBS2*, respectively. In vitro, SERPINF1 inhibits VEGF-A-induced motility and tube formation of endothelial cells and therefore strongly impairs sprouting angiogenesis [51]. *THBS2* encodes a protein that mediates cell–cell and cell–matrix interactions. Together with THBS1 and SERPINF1, it has also been shown to inhibit angiogenesis in patients with intrahepatic cholangiocarcinoma [52]. It is conceivable that the increased activity of SERPINF1 and THBS2 together with the high activity of HIF1A contribute to favoring of intussusceptive angiogenesis over conventional sprouting angiogenesis in the hypoxic environment of COVID-19 pulmonary involvement [12,49].

As expected, we found increased activity of genes coding for several collagen types both on the mRNA and on the protein level. One outstanding subtype was COL1, which showed a considerable increase in the late compared to the early group, especially in the adventitial layers of pulmonary artery branches and in the perivascular stromal tissue. One might hypothesize that the multitude of (micro-) thrombi in COVID-19 leads to elevated pulmonary vascular resistance and a resulting increase in pulmonary arterial pressure. Progressive perivascular fibrotic remodeling might therefore serve to strengthen the vessel walls in response to elevated intravascular pressures, serving as a nidus for uncontrolled pro-fibrotic remodeling in the lung due to the impaired microenvironment in COVID-19. An up-regulation in the late compared to the early group (and to controls, respectively) was also found for COL3 and COL4, predominantly in the internal elastic laminae of the pulmonary artery branches and in the alveolar basement membranes as well as for COL5 and COL6 in the media and adventitial layer of the pulmonary artery branches and the surrounding perivascular stromal tissue. Of note, the overall number of structures positive for these five collagen subtypes was significantly increased in the late group due to progressive fibrotic remodeling as shown in Appendix A.

Our study is limited by the small samples size and the use of autopsy samples. However, given the mostly large differences in group means and comparably small standard deviations, we thus have high confidence in the differentially expressed genes identified in our study. The autopsy tissue sample quality is generally limited due to the prolonged ischemia time before workup, resulting in a higher amount of autolysis and restricting the validity of gene expression analyses. However, the selection of tissue samples was performed with the utmost care to provide the best possible tissue quality and two well-matched cohorts to minimize diagnostic pitfalls. The individual viral loads have not been determined as these were out of the scope of this manuscript. Given that most patients of this study were treated with mechanical ventilation, it has to be considered that mechanical ventilation itself can set inflammatory and fibrogenic stimuli, particularly in the context of infection-related acute respiratory distress syndrome. Moreover, given that all patients died within three weeks after hospitalization, our results particularly reflect the acute and subacute disease phase while long-term fibrogenesis in reconvalescents was not within the scope of this study.

We could demonstrate that there is a shift from inflammation to connective tissue remodeling with increasing hospitalization time in patients with severe COVID-19 on both the gene expression and the protein level. In addition, this work provides evidence for IL1R1 and PDGFRB as potential therapeutic targets in future studies and reinforces the use of the interleukin 1 receptor antagonist anakinra. Future studies will have to prove the use of IL1R1 and PDGFRB as potential therapeutic targets via functional assays. For the first time, we could show a dynamic transition from inflammation and intussusceptive angiogenesis to angiogenesis-mediated fibrogenesis in COVID-19.

## 4. Materials and Methods

### 4.1. Sample Recruitment

Since the beginning of the worldwide SARS-CoV-2 pandemic, the German Registry of COVID-19 Autopsies (www.DeRegCOVID.ukaachen.de, accessed on 27 January 2022) has been established enabling tissue-based COVID-19 research. For this study, samples of this registry were used as well as samples from an autopsy series in Basel, Switzerland. To avoid misinterpretations, only patients without pre-existing pulmonary diseases were included. Formalin-fixed paraffin-embedded (FFPE) lung tissue from 12 patients has been extracted and assigned to two groups. For the “early group” we selected n = 6 patients who were hospitalized due to COVID-19 and deceased within the first seven days of hospitalization while n = 6 patients who were hospitalized for the same reason but deceased at least seven days after admission were assigned to the “late group”. For the comparative analysis n = 11 age and gender-matched downsizing lung tissue samples were used as healthy controls, i.e., tissue from donor lungs that was not used for lung transplantation due to surgical and procedural reasons. 

All patients or their relatives provided written informed consent for the use of their data and samples obtained during autopsy for scientific purposes. Ethical approval was given by the local institutional review board at Hannover Medical School (no. 6921_BO_K_2021 and no. 2702–2015).

### 4.2. mRNA Expression Analysis

One representative block of FFPE lung tissue with typical COVID-19 histopathology was selected from each patient and RNA was isolated using the Maxwell 16 LEV RNA FFPE Purification Kit (Promega Corporation, Madison, WI, USA). RNA content was assessed using the Qubit RNA IQ Assay (Thermo Fisher Scientific, Waltham, MA, USA) guaranteeing a minimum of 200 ng of RNA in each analyzed sample. Samples were analyzed using the Nanostring^®^ Fibrosis V2 panel including 770 human fibrosis-related target genes and the nCounter Analysis System (NanoString Technologies, Seattle, WA, USA) which is optimized for FFPE-based experiments. Normalization of counts was performed using the nSolver analysis software version 3.0 (NanoString Technologies, Seattle, WA, USA). Established housekeeping genes (glucuronidase beta (*GUSB*) and phosphoglycerate kinase 1 (*PGK1*)) were designated as reference genes for standardization of measurements. Further analyses on the ascertained log2 mRNA counts were performed using R software version 3.4.4 (R Foundation for Statistical Computing, Vienna, Austria) and the nCounter Advanced Analysis module version 1.1.5. The absolute gene expression values for all COVID-19 samples were analyzed and compared with those of the healthy control samples. Ultimately, samples from the “early group” were compared with those from the “late group”. U-tests were used for pairwise comparisons between groups, and Kruskal-Wallis tests were used for multigroup comparisons. Holm-Bonferroni corrected p-values (for gene expression) and false discovery rate (FDR; for pathway analyses) values <0.05 were considered statistically significant. Biological pathway analysis was performed using the GeneOntology database and gene-pathway associations supplied by Nanostring database.

### 4.3. Protein Expression Analysis

Immunohistochemistry (IHC) was performed on 2 µm thick slices of FFPE tissue. After de-paraffinization with xylene for 10 min twice, rehydration via decreasing concentration of ethanol was performed. The sections were subjected to heat-induced epitope retrieval in respective antibody buffer and stained with primary antibodies using the manufacturer’s protocol of ZytoChem Plus HRP Polymer Kit (Zytomed systems) with DAB (3, 3′-Diaminobinzidine) solution (the detailed antibody list can be found in Appendix A). After the staining Eukitt^®^ mounting media was used as an adhesive and a sealant. Images were taken after automated whole-slide imaging using the APERIO CS2 scanner (Leica Biosystems, Wetzlar, Germany) and ImageScope software Version 12.3.3.5048 (Leica Biosystems, Wetzlar, Germany).

## Figures and Tables

**Figure 1 ijms-23-01583-f001:**
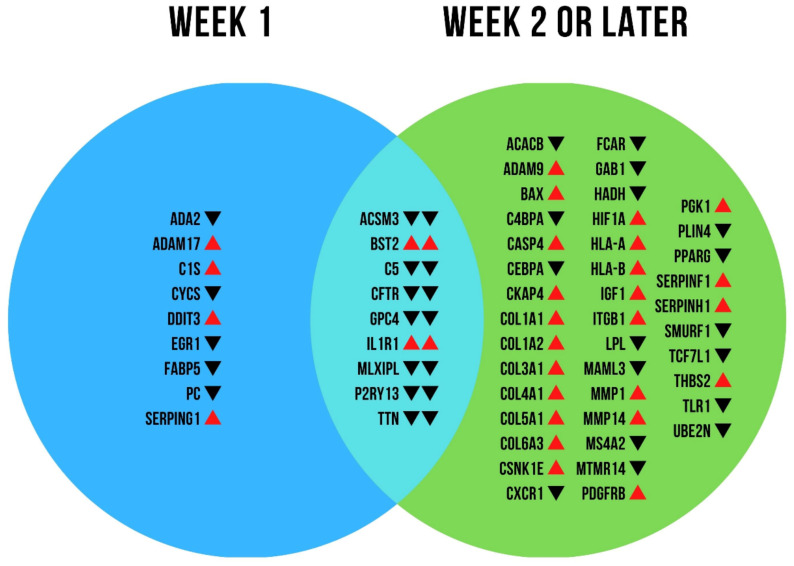
mRNA expression analysis. Venn diagram of differentially regulated genes over time. Arrows indicate increased (red arrow up) or decreased (black arrow down) activity of the respective genes in the group of patients who succumbed to COVID-19 within the first week of hospitalization (“week 1”) or later (“week 2 or later”) compared to healthy control lungs, respectively. In the overlapping area (middle), the left and right arrows indicate the expression in week 1 and week 2 or later, respectively.

**Figure 2 ijms-23-01583-f002:**
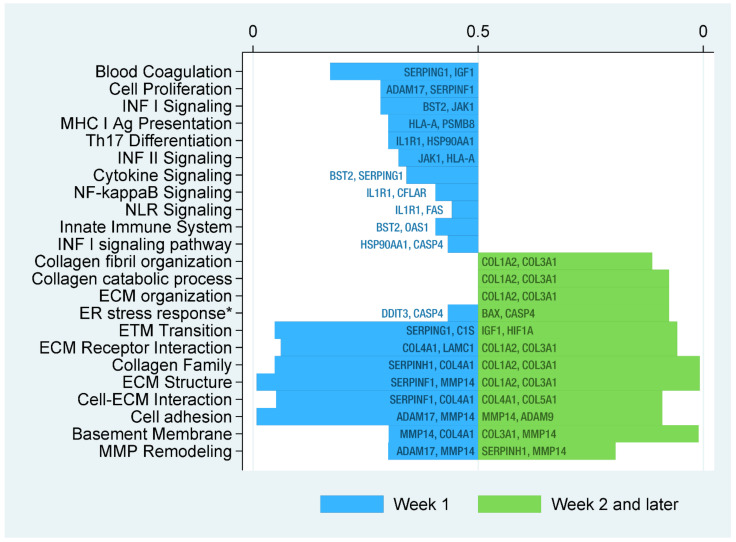
Significantly regulated biological pathways over time and corresponding fdr values. Functional pathway analysis using gene–pathway associations supplied by Nanostring and the GeneOntology database revealed an up-regulation of pro-inflammatory pathways in the early course of the disease, whereas fibrogenic pathways dominated on prolonged hospitalization (significance level, fdr < 0.05). FDR, false discovery rate; INF, interferon; MHC, major histocompatibility complex; NF, nuclear factor; NLR, NOD-like receptor; ECM, extracellular matrix; ER, endoplasmic reticulum; ETM, endothelial to mesenchymal; MMP, matrix metallopeptidase; * intrinsic apoptotic signaling pathway in response to endoplasmic reticulum stress.

**Figure 3 ijms-23-01583-f003:**
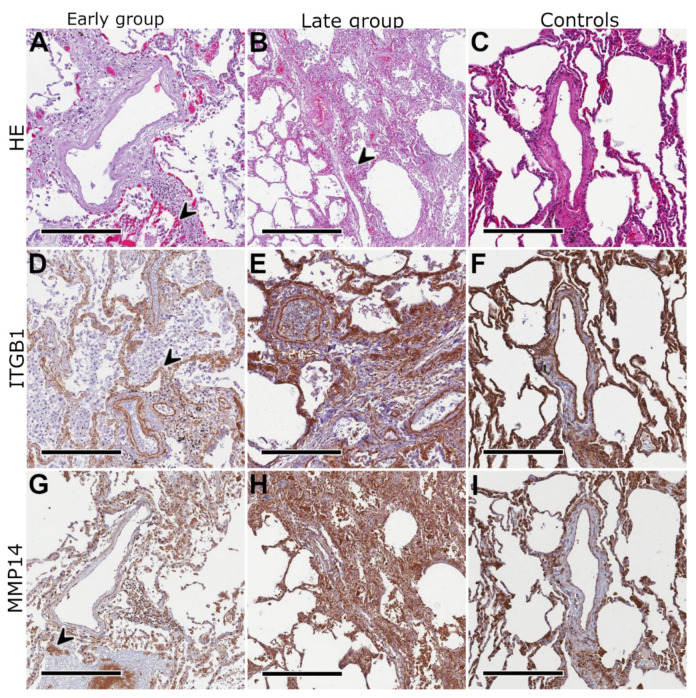
Immunostaining I. Exemplary histological morphology using hematoxylin-eosin (HE) staining in a patient who succumbed to COVID-19 3 days after hospital admission (**A**) showing a perivascular lymphocytic infiltrate and focal hyaline membranes/intraalveolar fibrin deposits (arrow) in line with diffuse alveolar damage; to compare, (**B**) shows another patient who died 14 days after hospitalization due to COVID-19 with intra-alveolar mesenchymal proliferation (arrow) and adjacent thickened alveolar septae indicating a pattern of acute fibrinous organizing pneumonia; (**C**) healthy lung tissue that was donated, but not used for lung transplantation; integrin subunit beta 1 (ITGB1) showed a specific but moderate staining of the internal and external elastic lamina of pulmonary artery branches as well as alveolar basement membranes (arrow) in the early group (**D**); in the late group, ITGB1 was enhanced in the same structures but in comparison to (**E**) these structures appeared distinctly thickened resulting in a higher staining intensity; matrix metalloproteinase 14 (MMP14) showed a broad but moderate staining of several extracellular matrix (ECM) structures in the early group (**G**) compared with an extensive high intensity enrichment in ECM structures in the late group (**H**); compared to the autopsy samples, staining appeared more intensive in healthy controls (**C**,**F**,**I**), presumably due to the higher tissue integrity; scale bars equal 300 µm in all panels except for (**B**) and (**H**) (500 µm each).

**Figure 4 ijms-23-01583-f004:**
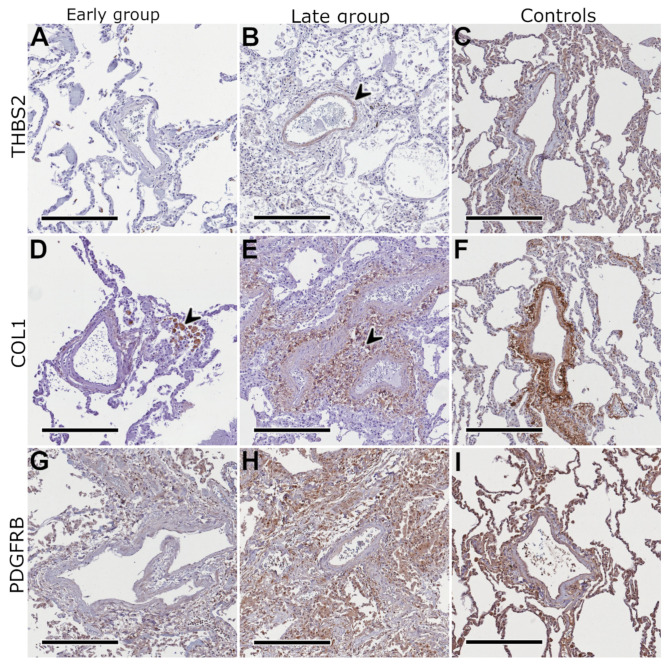
Immunostaining II. Thrombospondin 2 (THBS2) showed no enhancement in the early group (**A**) and a specific but slight enhancement in the media layer of pulmonary artery branches in the late group (**B**); collagen 1 (COL1) was enriched in macrophages (arrow) and minimally also in the perivascular stromal tissue in the early group (**D**) compared to a highly specific and powerful staining of the considerably augmented perivascular stromal tissue (arrow) in the late group (**E**); platelet-derived growth factor receptor beta (PDGFRB) showed a diffuse low-intensity enhancement in macrophages and in the walls of pulmonary artery branches in the early group (**G**) compared to an extensive enhancement predominantly in macrophages and ECM structures in the late group (**H**); compared to the autopsy samples, staining appeared more intensive in healthy controls (**C**,**F**,**I**), presumably due to the higher tissue integrity; scale bars equal 300 µm in all panels.

**Figure 5 ijms-23-01583-f005:**
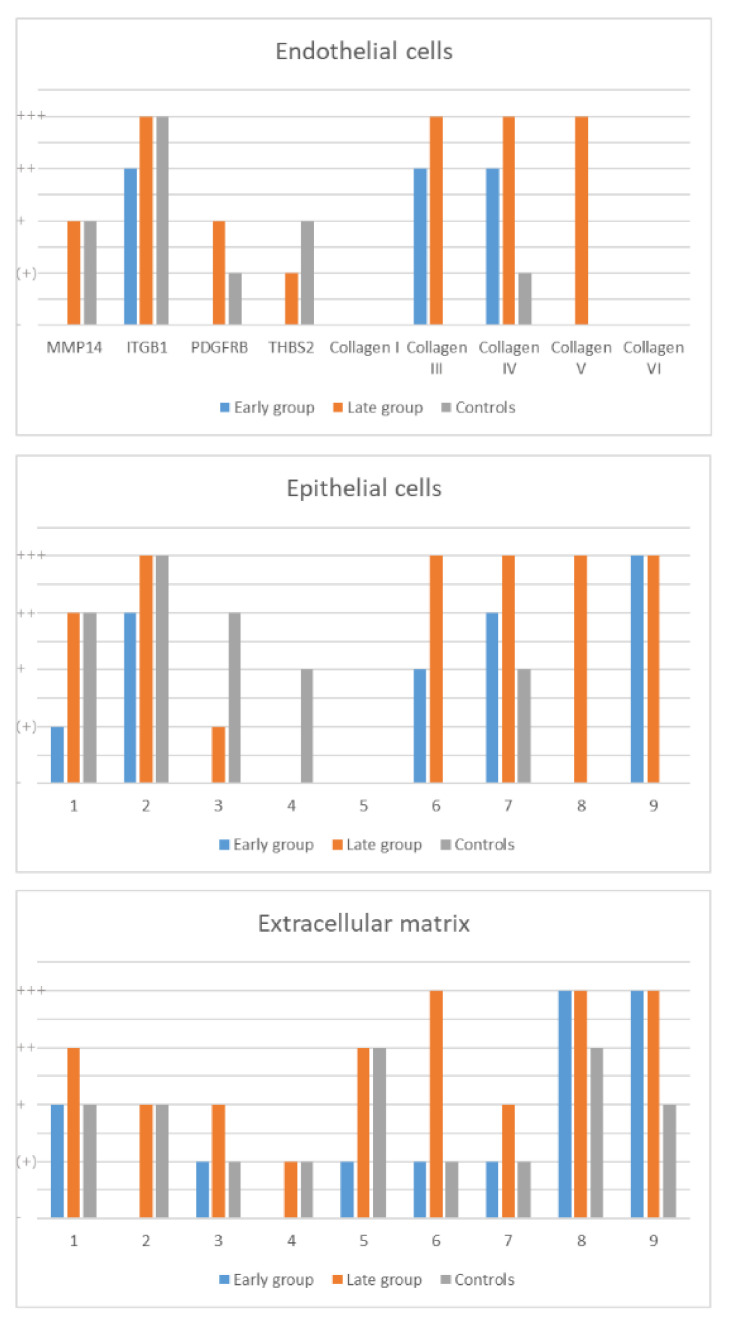
Semiquantitative analysis of the immunostaining. MMP14, matrix metalloproteinase 14; ITGB1, integrin beta type 1; PDGFRB, platelet derived growth factor receptor beta; THBS2, thrombospondin 2; +++, highly positive; ++, intermediate positive; +, slightly positive; (+), partially positive; -, negative.

## Data Availability

Data is contained within the Appendix A. The data presented in this study are available in Appendix A.

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
