# Peer review of "Time-Dependent Molecular Motifs of Pulmonary Fibrogenesis in COVID-19"

_ijms, 2022, doi:10.3390/ijms23031583_

Round 1

Reviewer 1 Report

The manuscript by Kamp et al describes the time dependent molecular events leading to pulmonary fibrogenesis in 12 Covid19 patient samples. They examined the mRNA and protein of expression as well as biological pathway analysis leading to development of pulmonary fibrosis. of 770 genes analysed, 58 genes exhibited an altered gene expression in Covid 19 patient samples compared to controls. Of all the samples tested, BST2 and IL1R1 displayed the highest expression independent of hospitalisation. Early hospitalisation revealed genes related to inflammation while late hospitalisation revealed genes related to fibrosis. Thereafter, they show that post first week of hospitalisation, there is a shift from inflammation to fibrosis in severe Covid patients. The authors finally conclude that IL1R1 and PDGFRB may represent therapeutics targets.

The manuscript is well written, however my concerns are outlined below

  1. The sample size is too small for definitive conclusion. 
  2. How did the authors define hospitalisation parameters for early and late group?
  3. Most of patients were on mechanical ventilation. Mechanical ventilation can significantly induce alteration in gene expression. How can authors validate these results?
  4. Why the authors explicitly studied ITGB!, MMP14, THBS2, COL1 and PDGFRB.
  5. The authors have mentioned that they have performed staining of healthy control samples. They must include in the manuscript to facilitate comparison.
  6. Fig 3, can the authors quantify to provide bar graphs?
  7. Data for Col 3,4,5,6 were not shown but results were mentioned.
  8. mRNA and protein levels of interested genes not shown. Only immunostaining was provided.
  9. The shift from inflammation to fibrosis must be explained in detail.
  10. Why IL1R1 and PDGFRB were increased throughout independent of hospitalisation?
  11. The conclusion that IL1R1 and PDGFRB can serve as therapeutic targets, must be supported by functional assays.
  12. Since the tissue samples are available, the authors can validate the results by WB.
  13. Did the patients have any comorbidites?

Author Response

We thank Reviewer 2 for the appreciative comments. We have addressed all his/her comments below and provided explanations where information was requested.

  • The introduction is descriptive, and authors should rewrite it to focus on giving the readers a brief about the alteration in genes expression in case of COVID-19 infection instead of history and clinical symptoms of COVID-19 infection.

 We thank reviewer #2 for this helpful comment. We revised the introduction according to his/her suggestions and    added more information on previous data concerning the altered gene expression in COVID-19 (lines 83-96).

  • The number of patients in both the early and late groups should be increased.

We thank the reviewer for this comment. Indeed, the sample size is relatively small. However, the amount of available material from COVID-19 patients is very limited. Actually, we used a large German registry of COVID-19 autopsy samples (DeRegCOVID.ukaachen.de), but there were three factors limiting the sample size. First, patients with preexisting pulmonary diseases prior to SARS-CoV2 infection were excluded from the study. Second, hospitalization time before death was documented solely for n=16 patients (n=8 for both, < and >7 days of hospitalization before death). Third, as discussed in the limitations paragraph of our manuscript, RNA quality in autopsy samples is also limited due to autolysis and FFPE preparation. We isolated RNA from all 16 cases and selected the best 6 cases of each group to guarantee the best possible mRNA expression results. Given these requirements, we believe that n=6/group is a considerable number of patients. Additionally, the mRNA expression results, as well as the histomorphological analysis, showed a relatively homogenous pattern in each group, reinforcing the suitability of both cohorts for this kind of analysis. Moreover, regarding previous studies such as the initial description of morphological and molecular features of COVID-19 (Ackermann et al., NEJM, 2020, doi: 10.1056/NEJMoa2015432), comparable sample sizes have been shown to yield reliable results. In addition, regarding the statistical methods used in our study, solely Holm-Bonferroni corrected genes with a false discovery rate <0.05 were considered statistically significant, meaning a conservative assessment. Also, with the given minimum of 6 samples per group and aiming at a statistical power of 80% (no type II errors) we can assume any effect size (the difference between group means) of 2 times the pooled standard deviation to be considered significant (includes correction for non-parametric testing). Given the mostly large differences in group means and comparably small standard deviations, we thus have high confidence in the differentially expressed genes identified in this study. We added this to our revised manuscript in lines 373-376.

  • In the clinical information part, it is extremely important to indicate if the patients included in the current study were suffering from other preexisting medical conditions and if there was any pulmonary involvement.

We thank reviewer #2 for this comment. Among all patients included in this study, only 2 patients, both out of the late cohort, had documented comorbidities. One patient suffered from moderate obesity and one patient had a documented abstinent nicotine dependence. No patients suffered from pulmonary comorbidities, particularly no fibrosing lung diseases. We added this information to the results section in lines 122-125.

  • It would be more informative if you can quantify the intensity of immunohistochemical

staining parameters and compare early and late groups and control.

We thank the reviewer for this essential comment. We added the semiquantitative interpretation of the immunohistochemical staining analysis to the supplementary table B2, as well as bar graphs to figure 5 in our revised manuscript.

  • Table 1 should be moved to the supplementary material section.

We thank reviewer #2 for this suggestion. We moved the table 1 to the supplementary material. It is now designated as table A1 in the revised version of our manuscript.

Reviewer 2 Report

In this study, the authors investigated the genes expression profiles incase of both early and late COVID-19 infection stages; they showed that after one week from infection there was a shift from pro-inflammatory to fibrogenic pathways activation. And confirmed some of the gene expression findings through immunohistochemical detection. They presented their findings in a very simple way and their findings are of importance in understanding the development of COVID infection cases. Here are some recommendations to the authors that might help in improving the quality of their manuscript.

Major:

1- The introduction is descriptive, and authors should rewrite it to focus on giving the readers a brief about the alteration in genes expression in case of COVID-19 infection instead of history and clinical symptoms of COVID-19 infection.

2- The number of patients in both the early and late groups should be increased.

3- In the clinical information part, it is extremely important to indicate if the patients included in the current study were suffering from other preexisting medical conditions and if there was any pulmonary involvement.

4- It would be more informative if you can quantify the intensity of immunohistochemical staining parameters and compare early and late groups and control.

Minor:
- Table 1 should be moved to the supplementary material section.

Author Response

We thank Reviewer #1 for the valuable comments which helped to significantly improve our manuscript. We have addressed all comments below and provided explanations where information was requested.

  • The sample size is too small for definitive conclusion.

We thank the reviewer for this comment. Indeed, the sample size is relatively small and therefore definitive conclusion is challenging. However, the amount of available high quality material from COVID-19 patients is also very limited. Actually, we relied on the rather large national German registry of COVID-19 autopsy samples (DeRegCOVID.ukaachen.de); however there were three factors limiting the sample size used. First, patients with preexisting pulmonary diseases prior to SARS-CoV-2 infection were excluded from the study (this point was added in lines 402-403 of our revised manuscript). Second, hospitalization time before death was documented solely for n=16 patients (n=8 for both, < and >7 days of hospitalization before death). Third, as discussed in the limitations paragraph of our manuscript, RNA quality in autopsy samples is also limited due to autolysis and FFPE preparation. We isolated RNA from all these 16 cases and selected the best 6 cases of each group to guarantee the best possible mRNA expression results. Given these requirements, we believe that n=6/group is a considerable number of patients overall. Additionally, the mRNA expression results as well as the histomorphological analysis showed a comparatively homogenous pattern in each group, reinforcing the suitability of both cohorts for our purposes. Moreover, regarding previous studies, such as the initial description of morphological and molecular features of COVID-19 (Ackermann et al., NEJM, 2020, doi: 10.1056/NEJMoa2015432), comparable sample sizes have been shown to yield reliable results. In addition, regarding the statistical methods used in our study, solely Holm-Bonferroni corrected genes with a false discovery rate <0.05 were considered as statistically significant, a quite conservative assessment. Also, with the given minimum of 6 samples per group and aiming at a statistical power of 80% (no type II errors), we can assume any effect size (the difference between group means) of 2 times the pooled standard deviation to be considered significant (includes correction for non-parametric testing). Given the mostly large differences in group means and comparably small standard deviations, we thus have high confidence in the differentially expressed genes identified in our study. We added this to our revised manuscript in lines 373-376.

  • How did the authors define hospitalization parameters for early and late group?

We thank the reviewer for this essential question. Most patients were admitted to the hospital due to progressive dyspnea but unfortunately, clinical information on further extra-pulmonary symptoms was not available.

  • Most of the patients were on mechanical ventilation. Mechanical ventilation can significantly induce alteration in gene expression. How can the authors validate these results?

We thank the reviewer for this critical comment. Indeed, mechanical ventilation can induce fibrotic remodeling that results in an altered gene expression signature. Hence, we can’t exclude definitely, that our results are at least slightly influenced by mechanical ventilation itself. However, patients who were mechanically ventilated and those who were not were included in this study and gene expression data was quite homogenous within each given group. Therefore, we believe that the influence of mechanical ventilation on gene expression might be of minor importance in this study. In addition, the sample size of this study was too small for subgroup analyses, so we could not directly address this issue. Additionally, in a previous study (Ackermann et al., NEJM, 2020), we performed a subgroup-analysis of patients who received mechanical ventilation and patients who did not in the context of COVID-19 and we could not demonstrate a significant difference between both groups with regard to the gene expression.

  • Why did the authors explicitly study ITGB1, MMP14, THBS2, COL1 and PDGFRB.

We thank reviewer #1 for this important question. We focused on genes that showed an increased expression compared to controls after week 2, but not earlier, to identify genes and proteins that might potentially contribute to the process of fibrogenesis in COVID-19. We tried our best to select a broad group of genes out of the n=23 genes that showed increased expression on the mRNA level in the late group for immunostaining, in order to provide the best possible evidence for the relevance of these genes/proteins in this context and therefore selected the 5 genes/proteins mentioned above, as well as further collagen subtypes.

  • The authors have mentioned that they have performed staining of healthy control samples. They must include in the manuscript to facilitate comparison.

We thank the reviewer for this relevant point. We added healthy controls to the immunostaining figure and divided it into 2 subfigures (figure 3 and figure 4 in the revised manuscript).

  • Fig 3, can the authors quantify to provide bar graphs?

We thank reviewer #1 again for this question. We added the semi-quantitative interpretation of the immunohistochemical staining analysis to the supplementary table B2 as well as bar graphs as figure 5 of the main document.

  • Data for Col 3,4,5,6 were not shown but results were mentioned.

We thank the reviewer for this comment. The immunostaining results of collagen subtypes 3, 4, 5, and 6 in all groups including healthy controls are depicted in the supplemental figure C1. We would prefer to leave them in the supplement in order not to overload the main manuscript.

  • mRNA and protein levels of interested genes not shown. Only immunostaining was provided.

We thank reviewer #1 for this suggestion. We added box plots of all genes with increased mRNA expression to supplementary figure A1 as well as a reference in lines 131-132. In addition, the raw date of the mRNA expression are deposited in the supplementary table C1.

  • The shift from inflammation to fibrosis must be explained in detail.

We thank reviewer #1 again for this critical comment. We revised the respective paragraph (lines 164-173) and added a more precise description of the shift from inflammation to fibrosis. In addition, we created a flow plot to clarify the shift (supplementary table A2) and supplied the respective data in supplementary table A3.

  • Why IL1R1 and PDGFRB were increased throughout independent of hospitalization?

We thank the reviewer for this question. Indeed, all patients analyzed in this study had already been hospitalized and succumbed to COVID-19 during the hospital stay. In both, patients who deceased within the first week as well as patients who deceased later, increased expression was found for IL1R1 and PDGFRB compared to healthy controls. Thus, we concluded that IL1R1 and PDGFRB might represent key molecules of the inflammatory process and fibrogenesis in COVID-19 respectively.

  • The conclusion that IL1R1 and PDGFRB can serve as therapeutic targets, must be supported by functional assays.

We thank the reviewer for this comment and we agree with the suggestion concerning the need for functional assays to assess the suitability of IL1R1 and PDGFRB as potential molecular targets in COVID-19. However, these assays have to be performed under safety level S3 conditions due to the risk of infections and respecting several legal regulations: human autopsy samples from patients who succumbed to COVID-19 have to be formaldehyde-fixed for at least 3 days before work-up spoiling functional assays; future works will have to approach this issue. We included this relevant point in the limitations/perspectives section of our revised discussion in lines 392-393.

  • Since the tissue samples are available, the authors can validate the results by WB.

We thank reviewer #1 for this essential suggestion. Western blotting indeed represents an appropriate method for confirmatory testing of the protein expression based on the results of the gene expression analysis. However, this method requires high quality tissue samples such as fresh human lung tissue or cryo-samples, while Western blotting from formalin-fixed paraffin-embedded material is very challenging. In addition, immunohistochemical staining, interpreted by experienced pathologists, represents the gold-standard in human pathology and additionally provides compartment-specific analysis of the protein expression. Therefore, we decided not to use Western blotting

  • Did the patients have any comorbidities?

We thank reviewer #1 for this question. Indeed, among all patients included in this study, only 2 patients, both out of the late cohort, had documented comorbidities. One patient suffered from moderate obesity and one patient had a documented abstinent nicotine dependence. No patients suffered from pulmonary comorbidities, particularly no fibrosing lung diseases. We added this information to the results section in lines 122-125.

Round 2

Reviewer 1 Report

The authors addressed all my concerns and the manuscript is significantly improved.